# TACKLING OVERSMOOTHING OF GNNS WITH CONTRASTIVE LEARNING

## ABSTRACT

Graph neural networks (GNNs) integrate the comprehensive relation of graph data and the representation learning capability of neural networks, which is one of the most popular deep learning methods and achieves state-of-the-art performance in many applications, such as natural language processing and computer vision. In real-world scenarios, increasing the depth (i.e., the number of layers) of GNNs is sometimes necessary to capture more latent knowledge of the input data to mitigate the uncertainty caused by missing values. However, involving more complex structures and more parameters will decrease the performance of GNN models. One reason called oversmoothing is recently introduced but the relevant research remains nascent. In general, oversmoothing makes the final representations of nodes indiscriminative, thus deteriorating the node classification and link prediction performance. In this paper, we first survey the current de-oversmoothing methods and propose three major metrics to evaluate a de-oversmoothing method, i.e., constant divergence indicator, easy-to-determine divergence indicator, and model-agnostic strategy. Then, we propose the Topology-guided Graph Contrastive Layer, named TGCL, which is the first de-oversmoothing method maintaining all three mentioned metrics. With the contrastive learning manner, we provide the theoretical analysis of the effectiveness of the proposed TGCL. Last but not least, we design extensive experiments to illustrate the empirical performance of TGCL comparing with state-of-the-art baselines.

## 1 INTRODUCTION

Combining the graph data comprehensive relations with the neural network models representation learning ability, graph neural networks (GNNs) achieve state-of-the-art performances in many real-world applications, such as document classification, natural language processing, computer vision, and recommender systems (Zhang et al., 2019). GNNs consist of many variant neural network models with different message-passing mechanisms, to name a few, such as GCN (Kipf & Welling, 2017), GraphSAGE (Hamilton et al., 2017), GAT (Velickovic et al., 2018), GIN (Xu et al., 2019), and GMNN (Qu et al., 2019).

In the complex real-world settings of applying GNNs, not every node is lucky enough to have node labels and/or node features. Hence, increasing the depth (i.e., the number of layers) of GNNs is a viable solution to capture more latent knowledge to reduce the uncertainty caused by missing values (Zhao & Akoglu, 2020). However, as the number of layers increases, the performance of GNN will decrease to a large degree (Kipf & Welling, 2017). The reasons may come from many aspects after involving more parameters like vanishing gradient, overfitting, and oversmoothing. Compared with the first two reasons, oversmoothing of GNNs is recently introduced (Li et al., 2018; Oono & Suzuki, 2020) and widely discussed (Chen et al., 2020a; Zhao & Akoglu, 2020; Rong et al., 2020; Chen et al., 2020b; Liu et al., 2020; Zhou et al., 2020). It is the phenomenon that the learned node representations become indistinguishable as the number of the hidden layers increases, thus hurting the performance of down-streaming tasks like node classification and link prediction.

To tackle the oversmoothing problem of GNNs, some nascent research works are proposed (Klicpera et al., 2019; Chen et al., 2020a; Zhao & Akoglu, 2020; Rong et al., 2020; Chen et al., 2020b; Liu et al., 2020; Zhou et al., 2020). They share the same logic (i.e., keeping the divergence between nodes) but differ in specific methodologies (i.e., rescaling divergences of learned representa-

tions (Zhao & Akoglu, 2020), adding the divergence regularizer in the learning process (Chen et al., 2020a; Zhou et al., 2020), changing input graph structures (Chen et al., 2020a; Rong et al., 2020; Chen et al., 2020b), or personalizing the information aggregation for each specific node (Klicpera et al., 2019; Liu et al., 2020)). Despite of good performance, some drawbacks still exist in those mentioned solutions. By surveying these SOTA de-oversmoothing strategies, we summarize three major metrics to evaluate a de-oversmoothing strategy: 1) *constant divergence indicator*, 2) *easy-to-determine divergence indicator*, and 3) *model-agnostic de-oversmoothing strategy*. (The detailed discussion could be found in Section 2). We find that no prevalent de-oversmoothing methods for GNNs could maintain all of them.

To bridge this gap, we propose a Topology-guided Graph Contrastive Layer (TGCL) with the inspiration from the contrastive learning concept (van den Oord et al., 2018), where we contrast node topological information to obtain discriminative node representations after many GNN layers. TGCL is the first de-oversmoothing strategy attempting to maintain all three mentioned metrics. Specifically, we set a constant and easy-to-determine divergence indicator between nodes, which is purely based on the topology of the input graph. With this divergence indicator, we aim to guide latent representations of neighbor node pairs closer and non-neighbor node pairs farther apart to mitigate the oversmoothing of GNNs. Last but not least, the proposed TGCL is model-agnostic, which means TGCL could be incorporated in multiple GNN models. With theoretical proof and empirical analysis, we show that the proposed TGCL could alleviate the oversmoothing problem of GNNs to a large extent.

Our contributions can be summarized as follows:

- We survey current de-oversmoothing methods by analyzing the advantages and the disadvantages of each method and summarize three metrics to evaluate a de-oversmoothing method for GNNs.
- We propose a topology-guided graph contrastive layer named TGCL to tackle the oversmoothing problem of GNNs, which enjoys all three metrics simultaneously.
- We show the effectiveness of the proposed TGCL from the theoretical proof and the empirical aspect with extensive experiments.

The rest of this paper is organized as follows. After a brief survey of de-oversmoothing methods in Section 2, we introduce the proposed TGCL in Section 3. The empirical evaluation of the proposed TGCL on real-world datasets is presented in Section 4. Then, we review the related work in Section 5 before we conclude the paper in Section 6.

## 2 BACKGROUND

As mentioned above, de-oversmoothing methods (Klicpera et al., 2019; Chen et al., 2020a; Zhao & Akoglu, 2020; Rong et al., 2020; Chen et al., 2020b; Liu et al., 2020; Zhou et al., 2020) share the same logic of keeping the divergence between node representations but differ in specific methodologies focusing on different merits. By taking the union of the metrics used in different state-of-the-arts, we get three metrics to evaluate a de-oversmoothing algorithm comprehensively.

There are three metrics as shown in Table 1, including constant divergence indicator, easy-to-determine divergence indicator, and model-agnostic de-oversmoothing strategy. Divergence indicator is indispensable for guiding the final node representation similarity based on the specified distance measurement. Several de-oversmoothing methods like (Klicpera et al., 2019; Zhao & Akoglu, 2020; Chen et al., 2020b; Liu et al., 2020) achieve the constant divergence indicator, which means the guidance is much more robust and not dependent on the training process of GNNs. However, to guide the node representation similarity reasonably, the divergence indicator is not that easy to be determined. For example, PairNorm (Zhao & Akoglu, 2020) is proposed as a normalization layer to keep the divergence of node representation against the original node feature. Instead of adding this regularizer directly to the learning objective of GNN models, PairNorm takes an alternative by rescaling the learned node representations with a constant hyperparameter to keep the original node feature divergence. PairNorm achieves two metrics: constant divergence indicator (i.e., the constant hyperparameter) and model-agnostic strategy (i.e., PairNorm can be added on different GNN models as a layer). However, the selection of that constant hyperparameter heavily depends

Table 1: Comparison of current de-oversmoothing methods

| | Constant Divergence Indicator | Easy-to-Determine Divergence Indicator | Model-Agnostic Strategy |
|---|---|---|---|
| APPNP (Klicpera et al., 2019) | Yes | No | Yes |
| MADReg + AdaEdge (Chen et al., 2020a) | No | Not Sure | Yes |
| PairNorm (Zhao & Akoglu, 2020) | Yes | No | Yes |
| DropEdge (Rong et al., 2020) | No | No | Yes |
| GCNII (Chen et al., 2020b) | Yes | Yes | No |
| DAGNN (Liu et al., 2020) | Yes | No | Yes |
| DGN (Zhou et al., 2020) | No | No | Yes |
| TGCL (Our Method) | Yes | Yes | Yes |

on the prior knowledge of the input graph data, which is hard to determine. (The discussion of other de-oversmoothing methods can be found in Section 5.)

As shown in Table 1, PairNorm is an effective de-oversmoothing method that maintains two metrics but needs prior knowledge to scale divergence between node pairs. While our proposed TGCL transfers this hard-to-acquire prior knowledge into the topology information of the input graph, where the divergence guidance between nodes is constant and easy to be determined. To be specific, our TGCL is the first de-oversmoothing method attempting to maintain these three metrics at the same time. In the next section, we formally introduce the proposed TGCL with theoretical proof for the model effectiveness. Moreover, we prove that the objective of PairNorm is just a special case of our TGCL, which shows the effectiveness of our TGCL from another perspective.

## 3 PROPOSED METHOD

In this section, we begin with the notions used in this paper. Then, we prove that the objective of the de-oversmoothing model PairNorm (Zhao & Akoglu, 2020) is a just special case of our Topology-guided Graph Contrastive Layer (TGCL). After analyzing the limitations of PairNorm, we formally introduce our proposed TGCL and show why it could better alleviate the oversmoothing issue with the contrastive learning manner.

### 3.1 NOTION

Throughout this paper, we use regular letters to denote scalars (*e.g.*, $\alpha$), boldface lowercase letters to denote vectors (*e.g.*, $\boldsymbol{v}$), and boldface uppercase letters to denote matrices (*e.g.*, $\boldsymbol{A}$). We formalize the graph mining problem in the context of undirected graph $\mathcal{G} = (\boldsymbol{V}, \boldsymbol{E}, \boldsymbol{X})$, where $\boldsymbol{V}$ consists of $n$ vertices, $\boldsymbol{E}$ consists of $m$ edges, $\boldsymbol{X} \in \mathbb{R}^{n \times d}$ denote the feature matrix and $d$ is the feature dimension. We let $\boldsymbol{A} \in \mathbb{R}^{n \times n}$ denote the adjacency matrix, $\boldsymbol{D} \in \mathbb{R}^{n \times n}$ denote the diagonal matrix of vertex degrees, and $\boldsymbol{I} \in \mathbb{R}^{n \times n}$ denote the identity matrix. For ease explanation, we denote $\boldsymbol{v}_i$ as node $i$, $\boldsymbol{x}_i$ as the input feature of node $i$, $\boldsymbol{z}_i$ as the embedding of node $i$ by any type of GNNs and $\boldsymbol{A}_i$ as the adjacency vector for node $i$. $\mathcal{N}_i$ is a set that contains the neighbors of node $i$ and $\bar{\mathcal{N}}_i$ is the complement of $\mathcal{N}_i$, which contains the non-neighbor of node $i$.

### 3.2 PRELIMINARY

Each graph convolutional layer can be understood as a smoothing operation (Li et al., 2018) but stacking many layers renders the final representation of a node indistinguishable from others. Therefore, how to recover the divergence between node representations but preserving the shared information becomes a vital problem in graph mining. In PairNorm (Zhao & Akoglu, 2020), the divergence between node pairs is based on a hyper-parameter, which requires prior knowledge of the input graph data and is hard to acquire. More specifically, PairNorm is proposed as a novel normalization layer to prevent all node embeddings from becoming too similar by minimizing the following objective:

$$L_p = \sum_{\boldsymbol{v}_i \in V} \|\boldsymbol{z}_i - \boldsymbol{x}_i\|^2 + \sum_{(i,j) \in E} \|\boldsymbol{z}_i - \boldsymbol{z}_j\|^2 - \sum_{(i,k) \notin E} \|\boldsymbol{z}_i - \boldsymbol{z}_k\|^2 \qquad (1)$$

where $\boldsymbol{z}_i$ is the node embedding vector of node $\boldsymbol{v}_i$ and $\boldsymbol{x}_i$ is the original feature vector of node $\boldsymbol{v}_i$. In the equation above, the first term is the reconstruction error, the second term is responsible for min-

imizing the difference between two representations of a neighbor node pair, and the last term aims to maximize the difference between two representations of a remote node pair. By reformulating Eq. 1, we could derive a upper bound of $L_p$ in the form of contrastive learning loss term as follows:

$$
\begin{aligned}
L_p &= \sum_{v_i \in V} \|z_i - x_i\|^2 + \sum_{v_i \in V} \sum_{v_j \in \mathcal{N}_i} \|z_i - z_j\|^2 - \sum_{v_i \sim V} \sum_{v_k \notin \mathcal{N}_i} \|z_i - z_k\|^2 \\
&= \sum_{v_i \sim V} \|z_i - x_i\|^2 - \sum_{v_i \sim V} \sum_{v_j \in \mathcal{N}_i} \log(e^{-\|z_i - z_j\|^2}) + \sum_{v_i \sim V} \sum_{v_k \notin \mathcal{N}_i} \log(e^{-\|z_i - z_k\|^2}) \qquad (2) \\
&\leq \sum_{v_i \sim V} \|z_i - x_i\|^2 - \sum_{v_i \sim V} \sum_{v_j \in \mathcal{N}_i} \log(e^{-\|z_i - z_j\|^2}) + \sum_{v_i \sim V} \log(\sum_{v_k \notin \mathcal{N}_i} e^{-\|z_i - z_k\|^2}) \qquad (3) \\
&\leq \sum_{v_i \sim V} \|z_i - x_i\|^2 - \sum_{v_i \sim V} \sum_{v_j \in \mathcal{N}_i} \log(e^{-\|z_i - z_j\|^2}) + \sum_{v_i \sim V} \sum_{v_j \in \mathcal{N}_i} \log(\sum_{v_k \notin \mathcal{N}_i} e^{-\|z_i - z_k\|^2}) \\
&= \sum_{v_i \sim V} \|z_i - x_i\|^2 + \sum_{v_i \sim V} \sum_{v_j \in \mathcal{N}_i} \log(\frac{\sum_{v_k \notin \mathcal{N}_i} e^{-\|z_i - z_k\|^2}}{e^{-\|z_i - z_j\|^2}}) \\
&\leq \sum_{v_i \sim V} \|z_i - x_i\|^2 + \sum_{v_i \sim V} \sum_{v_j \in \mathcal{N}_i} [\log(1 + \frac{\sum_{v_k \notin \mathcal{N}_i} e^{-\|z_i - z_k\|^2}}{e^{-\|z_i - z_j\|^2}})] \\
&= \sum_{v_i \sim V} \|z_i - x_i\|^2 - \sum_{v_i \sim V} \sum_{v_j \in \mathcal{N}_i} [\log(\frac{e^{-\|z_i - z_j\|^2}}{e^{-\|z_i - z_j\|^2} + \sum_{v_k \notin \mathcal{N}_i} e^{-\|z_i - z_k\|^2}})] \\
&= \sum_{v_i \sim V} \|z_i - x_i\|^2 - \sum_{v_i \sim V} \sum_{v_j \in \mathcal{N}_i} [\log(\frac{f(z_i, z_j)}{f(z_i, z_j) + \sum_{v_k \notin \mathcal{N}_i} f(z_i, z_k)})] = L_1 \qquad (4)
\end{aligned}
$$

where $f(z_i, z_k) = e^{-\|z_i - z_k\|^2}$. Here, we apply Jensen's inequality to derive Eq. 3 as a upper bound of Eq.2 since $\log(\cdot)$ is concave. We observe that $L_1$ is a upper bound of PairNorm and we could interpret two regularization terms $\|z_i - z_j\|^2$ and $\|z_i - z_k\|^2$ of PairNorm as a special case of a contrastive learning loss term in $L_1$ by setting the similarity measurement function $f(z_i, z_k)$ to be $e^{-\|z_i - z_k\|^2}$.

However, both PairNorm (Eq. 1) and the upper bound of PairNorm (Eq. 4) only consider the first-order neighbor information but neglect the $\mathcal{K}$-hop neighbors information. For example, in a real-world scenario, we are given a remote pair $(v_k, v_i)$. It is highly possible that $v_k$ and $v_i$ have the similar representations, if they share the same label information. However, simply minimizing the third term of PairNorm (i.e., $-\|z_i - z_k\|^2$) will push $z_i$ away from $z_k$, resulting in sub-optimal solution. In addition, if we are given two remote pairs $(v_{k_1}, v_i)$ and $(v_{k_2}, v_i)$ such that node $v_{k_1}$ is far from node $v_i$ and node $v_{k_2}$ is near node $v_i$ (e.g., 2-hop neighbor), the weight imposed on these two remote pairs should be different as we expect that $z_{k_1}$ should be more different from $z_i$ than $z_{k_2}$ due to the topological information in the graph. However, PairNorm and $L_1$ (Eq. 4) assume that all unconnected node pairs ($z_i$ and $z_k$) have the same weight by setting the weights to be 1 for neighbor pairs and remote pairs. Therefore, if the $\mathcal{K}$-hop neighbors of $z_i$ share the same topological structure of $z_i$ or the same label information, pushing $z_i$ away from the representation of its $\mathcal{K}$-hop neighbors ($\mathcal{K} > 1$) and ignoring the different weights for different remote pairs will result in a sub-optimal solution. Motivated by these, we propose to utilize the similarity of two adjacency vectors of each node pair and embed the global topological structure information into the representation of each node such that GNNs can derive better discriminative representations for all nodes.

## 3.3 OVERVIEW OF TGCL

The structure of TGCL is shown in Figure 1. TGCL is model-agnostic, and it can be added before the final output layer of any GNN model. To recover the divergence between node representations, we first need to determine the divergence between different node pairs. In TGCL, we transfer this hard-to-acquire knowledge into the topology information of the input graph, which is a constant divergence indicator (i.e., not varying with the depth of GNNs), easy to obtain, and purely depends on the adjacency vector of each node. The main idea of TGCL is to encode the topological divergence

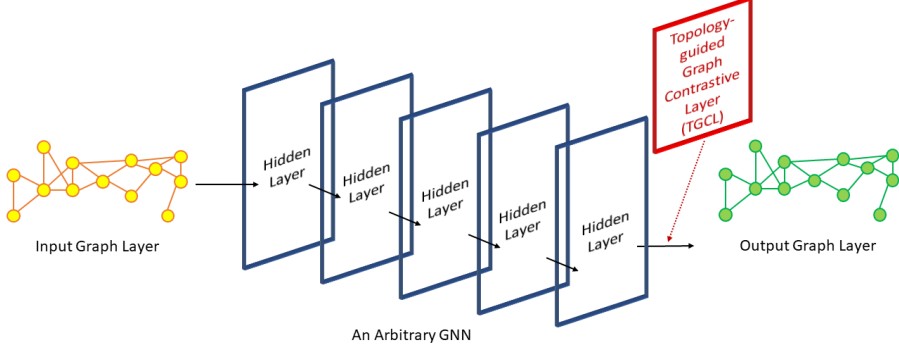

Figure 1: An arbitrary graph neural network with the proposed model-agnostic TGCL.

relationship of any pair of nodes into their final node representations. Specifically, we expect that the representations of two nodes are similar if their adjacency vectors are close enough. Otherwise, their representations should be discriminative. Thus, we propose the *topology-guided contrastive loss* formulated as follows:

$$L_{\text{TGCL}} = -\mathbb{E}_{\boldsymbol{v}_i \sim V} \mathbb{E}_{\boldsymbol{v}_j \in \mathcal{N}_i} [\log \frac{\sigma_{ij} f(\boldsymbol{z}_i, \boldsymbol{z}_j)}{\sigma_{ij} f(\boldsymbol{z}_i, \boldsymbol{z}_j) + \sum_{\boldsymbol{v}_k \in \bar{\mathcal{N}}_i} \gamma_{ik} f(\boldsymbol{z}_i, \boldsymbol{z}_k)}]$$

$$\sigma_{ij} = 1 - dist(\boldsymbol{A}_i, \boldsymbol{A}_j)/n, \ \ \gamma_{ik} = 1 + dist(\boldsymbol{A}_i, \boldsymbol{A}_k)/n \tag{5}$$

where $f(\cdot)$ is a similarity function, *e.g.*, $f(a, b) = \exp(\frac{a^T b}{\tau})$, $\tau$ is the temperature, $dist(\cdot)$ is a distance measurement function, *e.g.*, hamming distance, and the set $\bar{\mathcal{N}}_i$ contains the non-neighbor nodes of the node $i$. The intuition of Eq. 5 is that if $\boldsymbol{v}_i$ and $\boldsymbol{v}_j$ are neighbors, then the similarity of their representations should be as large as possible, while if $\boldsymbol{v}_i$ and $\boldsymbol{v}_k$ are two remote nodes (not connected in the graph), the similarity of their representations should be as small as possible, and the magnitude of dissimilarity is determined by how many neighbors these two nodes don't share. By adjusting the weights of both remote pairs and neighbor pairs based on the topological information, we aim to reduce the negative impact of remote nodes that have similar topological information.

To collaborate with different GNN models, the adaptive loss function $L_{total}$ is expressed as follows.

$$L_{total} = L_{agnostic} + \alpha L_{\text{TGCL}} \tag{6}$$

where $L_{agnostic}$ denotes the loss function of an arbitrary GNN model such as GCN (Kipf & Welling, 2017), and $\alpha$ is a constant hyperpararmeter and $L_{\text{TGCL}}$ stands for the loss function of our TGCL, which can serve as a regularizer to alleviate the over-smoothing problem.

### 3.4 THEORETICAL ANALYSIS OF TOPOLOGY-GUIDED CONTRASTIVE LOSS

In this subsection, we provide an analysis regarding the properties of the proposed contrastive loss.

**Lemma 1** *Given a neighbor node pair sampled from the graph $\mathcal{G} = (\boldsymbol{V}, \boldsymbol{E}, \boldsymbol{X})$, i.e., nodes $\boldsymbol{v}_i$ and $\boldsymbol{v}_j$, we have $I(\boldsymbol{z}_i, \boldsymbol{z}_j) \geq -L_{\text{TGCL}} + \mathbb{E}_{\boldsymbol{v}_i \sim V} \log(|\bar{\mathcal{N}}_i|)$, where $I(\boldsymbol{z}_i, \boldsymbol{z}_j)$ is the mutual information between two representations of the node pair $\boldsymbol{v}_i$ and $\boldsymbol{v}_j$, and $L_{\text{TGCL}}$ is the topology-guided contrastive loss weighted by hamming distance measurement.*

**Proof:** *Following the theoretical analysis in (van den Oord et al., 2018), the optimal value of $f(\boldsymbol{z}_i, \boldsymbol{z}_j)$ is given by $\frac{P(\boldsymbol{z}_j | \boldsymbol{z}_i)}{P(\boldsymbol{z}_j)}$. Thus, the weighted supervised contrastive loss could be rewritten as follows:*

$$L_{\text{TGCL}} = -\mathbb{E}_{\boldsymbol{v}_i \sim V} \mathbb{E}_{\boldsymbol{v}_j \in \mathcal{N}_i} [\log \frac{\sigma_{ij} f(\boldsymbol{z}_i, \boldsymbol{z}_j)}{\sigma_{ij} f(\boldsymbol{z}_i, \boldsymbol{z}_j) + \sum_{\boldsymbol{v}_k \in \bar{\mathcal{N}}_i} \gamma_{ik} f(\boldsymbol{z}_i, \boldsymbol{z}_k)}]$$

$$= \mathbb{E}_{\boldsymbol{v}_i \sim V} \mathbb{E}_{\boldsymbol{v}_j \in \mathcal{N}_i} [\log \frac{\sigma_{ij} \frac{P(\boldsymbol{z}_j | \boldsymbol{z}_i)}{P(\boldsymbol{z}_j)} + \sum_{\boldsymbol{v}_k \in \bar{\mathcal{N}}_i} \gamma_{ik} \frac{P(\boldsymbol{z}_k | \boldsymbol{z}_i)}{P(\boldsymbol{z}_k)}}{\sigma_{ij} \frac{P(\boldsymbol{z}_j | \boldsymbol{z}_i)}{P(\boldsymbol{z}_j)}}]$$

$$= \mathbb{E}_{\boldsymbol{v}_i \sim V} \mathbb{E}_{\boldsymbol{v}_j \in \mathcal{N}_i} [\log(1 + \frac{P(\boldsymbol{z}_j)}{\sigma_{ij} P(\boldsymbol{z}_j | \boldsymbol{z}_i)} \sum_{\boldsymbol{v}_k \in \bar{\mathcal{N}}_i} \gamma_{ik} \frac{P(\boldsymbol{z}_k | \boldsymbol{z}_i)}{P(\boldsymbol{z}_k)})]$$

*Since $(\boldsymbol{v}_i, \boldsymbol{v}_k)$ is defined as a remote node pair, it means that node $\boldsymbol{v}_i$ and node $\boldsymbol{v}_k$ are not connected in the graph, i.e., $\boldsymbol{A}(i,k) = A(k,i) = 0$. Therefore, we have $\gamma_{ik} \in (1, 2]$ for all remote nodes $\boldsymbol{v}_k$ and $\sigma_{ij} \in (0, 1]$ for all neighbor nodes $\boldsymbol{v}_j$ with hamming distance measurement, which leads to $\frac{1}{\sigma_{ij}} \cdot \frac{P(\boldsymbol{z}_j)}{P(\boldsymbol{z}_j|\boldsymbol{z}_i)} \geq \frac{P(\boldsymbol{z}_j)}{P(\boldsymbol{z}_j|\boldsymbol{z}_i)}$ and $\gamma_{ik} \frac{P(\boldsymbol{z}_k|\boldsymbol{z}_i)}{P(\boldsymbol{z}_k)} \geq \frac{P(\boldsymbol{z}_k|\boldsymbol{z}_i)}{P(\boldsymbol{z}_k)}$. Thus, we have*

$$L_{\text{TGCL}} \geq \mathbb{E}_{\boldsymbol{v}_i \sim V} \mathbb{E}_{\boldsymbol{v}_j \in \mathcal{N}_i} [\log(\frac{P(\boldsymbol{z}_j)}{P(\boldsymbol{z}_j|\boldsymbol{z}_i)} \sum_{\boldsymbol{v}_k \in \bar{\mathcal{N}}_i} \frac{P(\boldsymbol{z}_k|\boldsymbol{z}_i)}{P(\boldsymbol{z}_k)})]$$

$$\approx \mathbb{E}_{\boldsymbol{v}_i \sim V} \mathbb{E}_{\boldsymbol{v}_j \in \mathcal{N}_i} [\log(\frac{P(\boldsymbol{z}_j)}{P(\boldsymbol{z}_j|\boldsymbol{z}_i)} (|\bar{\mathcal{N}}_i| \mathbb{E}_{\boldsymbol{v}_k} \frac{P(\boldsymbol{z}_k|\boldsymbol{z}_i)}{P(\boldsymbol{z}_k)}))]$$

$$= \mathbb{E}_{\boldsymbol{v}_i \sim V} \mathbb{E}_{\boldsymbol{v}_j \in \mathcal{N}_i} [\log(\frac{P(\boldsymbol{z}_j)}{P(\boldsymbol{z}_j|\boldsymbol{z}_i)} |\bar{\mathcal{N}}_i|)]$$

$$\geq \mathbb{E}_{\boldsymbol{v}_i \sim V} \mathbb{E}_{\boldsymbol{v}_j \in \mathcal{N}_i} [\log(\frac{P(\boldsymbol{z}_j)}{P(\boldsymbol{z}_j|\boldsymbol{z}_i)}) + \log(|\bar{\mathcal{N}}_i|)]$$

$$= -I(\boldsymbol{z}_i, \boldsymbol{z}_j) + \mathbb{E}_{\boldsymbol{v}_i \sim V} \log(|\bar{\mathcal{N}}_i|)$$

*Finally, we have $I(\boldsymbol{z}_i, \boldsymbol{z}_j) \geq -L_{\text{TGCL}} + \mathbb{E}_{\boldsymbol{v}_i \sim V} \log(|\bar{\mathcal{N}}_i|)$, which completes the proof.*

Lemma 1 shows that the topology-guided contrastive loss for the graph is the lower bound of the mutual information between two representations of a neighbor node pair. Notice that $\mathbb{E}_{\boldsymbol{v}_i \sim V} \log(|\bar{\mathcal{N}}_i|)$ is the average logarithm of the number of unconnected edges for the nodes in the graph, which means that TGCL tends to be a better lower bound if imposed on a sparser graph.

## 4 EXPERIMENT

In this section, we demonstrate the performance of our proposed framework in terms of effectiveness by comparing it with state-of-the-art methods. In addition, we conduct a case study to show how the increase of the number of layers influences the performance of the proposed method.

### 4.1 EXPERIMENT SETUP

**Datasets:** Cora dataset is a citation network consisting of 2,708 scientific publications in seven classes and 5,429 edges. The edge in the graph represents the citation of one paper to another. CiteSeer dataset consists of 3,327 scientific publications which could be categorized into six classes and this citation network has 9,228 edges. PubMed is a diabetes dataset, which consists of 19,717 scientific publications in three classes and 88,651 edges. Reddit dataset is extracted from Reddit posts in September 2014, which consists of 4,584 nodes and 19,460 edges. In all experiments, we follow the splitting strategy used in (Zhao & Akoglu, 2020) by randomly sampling 3% of the nodes as the training samples, 10% of the nodes as the validation samples, and the rest 87% of the nodes as the test samples.

**Baselines:** We compared the performance of our method with the following baselines: (1) GCN (Kipf & Welling, 2017): vanilla graph convolutional network; (2) GCNII (Chen et al., 2020b): an extension of vanilla GCN with skip connections and additional identity matrices; (3) DGN (Zhou et al., 2020): the differentiable group normalization to normalize nodes within the same group and separate nodes among different groups; (4) PairNorm (Zhao & Akoglu, 2020): a novel normalization layer designed to prevent all node embeddings from becoming too similar; (5) DropEdge (Rong et al., 2020): a novel framework that randomly removes a certain number of edges from the input graph at each training epoch to reduce the speed of over-fitting and to prevent the oversmoothing issue. The reason why we do not include the de-oversmoothing strategies APPNP (Klicpera et al., 2019) and DAGNN (Liu et al., 2020) here is that they replace stacking layers with stacking hops propagation.

**Configuration:** In all experiments, we set the learning rate to be 0.0005 and the optimizer is Adam (Kingma & Ba, 2014). The feature dimension of the hidden layer is 50. The experiments are repeated 5 times if not specified. $dist(\cdot)$ is the hamming distance and $f(\cdot)$ is the cosine similarity measurement. All of the real-world datasets are publicly available. The code of our algorithms could

Table 2: Accuracy on node classification on four benchmark datasets.

| Method | Cora | CiteSeer | PubMed | Reddit |
|---|---|---|---|---|
| GCN | $0.6707 \pm 0.0519$ | $0.5578 \pm 0.0153$ | $0.7984 \pm 0.0077$ | $0.7537 \pm 0.0167$ |
| PairNorm | $0.7178 \pm 0.0064$ | $0.5628 \pm 0.0187$ | $0.7816 \pm 0.0087$ | $0.7592 \pm 0.0069$ |
| DropEdge | $0.7138 \pm 0.0186$ | $0.5330 \pm 0.0310$ | $0.8063 \pm 0.0128$ | $0.7539 \pm 0.0149$ |
| GCNII | $0.7179 \pm 0.0012$ | $0.5913 \pm 0.0050$ | $0.8035 \pm 0.0011$ | $0.7503 \pm 0.0068$ |
| DGN | $0.6896 \pm 0.0035$ | $0.5190 \pm 0.0141$ | $0.7929 \pm 0.0018$ | $0.7407 \pm 0.0321$ |
| TGCL | $0.7199 \pm 0.0151$ | $0.5783 \pm 0.0191$ | $0.8090 \pm 0.0065$ | $0.7556 \pm 0.0132$ |
| GCN+ResNet | $0.7453 \pm 0.0097$ | $0.6139 \pm 0.0197$ | $0.8127 \pm 0.0080$ | $0.7998 \pm 0.0181$ |
| PairNorm+ResNet | $0.7454 \pm 0.0327$ | $0.6054 \pm 0.0203$ | $0.8010 \pm 0.0086$ | $0.8040 \pm 0.0101$ |
| TGCL+ResNet | $0.7699 \pm 0.0113$ | $0.6125 \pm 0.0129$ | $0.8192 \pm 0.0013$ | $0.8106 \pm 0.0118$ |

be found in an anonymous link [*]. The experiments are performed on a Windows machine with a 16GB RTX 5000 GPU.

## 4.2 EXPERIMENTAL ANALYSIS

In this subsection, we evaluate the effectiveness of the proposed method on four benchmark datasets by comparing it with state-of-the-art methods. The base model for all methods we used in this experiment is graph convolutional neural network (GCN). For a fair comparison, we set the numbers of the hidden layers to be 10 for all methods and the dimension of the hidden layer to be 50. The experiments are repeated 5 times and we record the mean accuracy as well as the standard deviation in Table 2. By observation, we could find that our proposed method outperforms most baselines over these four datasets without adding ResNet. Though GCNII achieves the best performance in the CiteSeer dataset, it has worse performance in other datasets. When we further incorporate ResNet into the base model (GCN), PairNorm, and our proposed method, we observe the performance improvement for all methods. In addition, the gap of performance between GCN+ResNet and TGCL+ResNet becomes narrow. Our guess is that as we increase the number of layers, the vanishing gradient problem and the oversmoothing issue coexist in GCN based model. Adding ResNet into the base model somehow alleviates the issue of vanishing gradient, thus leading to great performance improvement for GNNs.

To further investigate the impact of oversmoothing issue, we conduct an experiment on the Cora dataset by increasing the number of layers. The x-axis of Figure 2 (a) is the number of layers and the y-axis is the accuracy on the test dataset. By observation, we find that without adding ResNet, the performance of GCN drops dramatically starting at 10 hidden layers. By comparing the performance of GCN and TGCL, we observe that after utilizing our proposed de-oversmoothing strategy, TGCL boosts the performance by more than 7.5% at 60 hidden layers. After adding ResNet, the performance of GCN+ResNet improves a lot due to the alleviation of the vanishing gradient problem, but we could still see more than 4% improvement at 60 hidden layers by our proposed method (TGCL+ResNet). Combining the experimental results in Table 2 and Figure 2 (a), we find that the oversmoothing issue slightly influences the performance of shallow GCN (when the number of layers is less than 10). However, GCN will suffer a lot from the oversmoothing issue, if we increase the number of layers to 20 or more (based on the results in Figure 2 (a)). To demonstrate the oversmoothing issue in other types of GNNs, we show the performance of our proposed with different base models (*e.g.*, GAT (Velickovic et al., 2018) and SGC (Wu et al., 2019)) in Figure 2 (b). The experimental setting is the same as the setting in Table 2. Figure 2 (b) shows that our proposed method outperforms GAT and SGC and thus alleviates the oversmoothing issue.

## 4.3 CASE STUDY: A MISSING FEATURE SCENARIO

*Why do we need a de-oversmoothing strategy, if increasing the number of the layers may result in a worse performance?* To answer this question, let's first imagine a scenario where some values of attributes are missing in the graph. In this scenario, the shallow GNNs may not work well because

---

[*]`https://drive.google.com/file/d/1t9_-5Hb35K7Vx7nD9is9-JmYJRn9Vz_M/ view?usp=sharing`

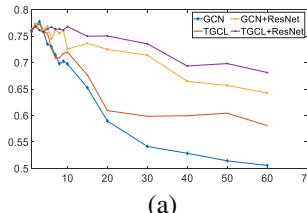 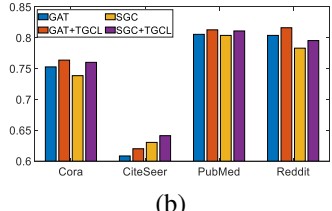 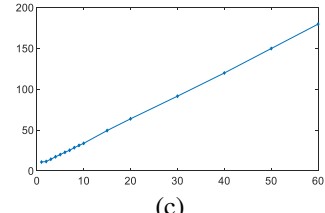

(a) (b) (c)

Figure 2: (a) accuracy vs the number of layers on Cora dataset; (b) accuracy of different base models on four datasets; (c) running time (in second) vs the number of layers on Cora dataset. (Best view in color)

Table 3: Accuracy of node classification on four datasets by masking $p$ percent of node attributes. $\#L$ denotes the number of layers where a model achieves the best performance.

| Node Feature Missing Rate | | $p = 25\%$ | | $p = 50\%$ | | $p = 75\%$ | |
|---|---|---|---|---|---|---|---|
| Dataset | Method | Acc | #L | Acc | #L | Acc | #L |
| Cora | GCN+ResNet | $0.731 \pm 0.009$ | 3 | $0.729 \pm 0.010$ | 11 | $0.688 \pm 0.018$ | 11 |
| | TGCL+ResNet | $0.732 \pm 0.010$ | 6 | $0.751 \pm 0.016$ | 15 | $0.717 \pm 0.031$ | 30 |
| Citeseer | GCN+ResNet | $0.615 \pm 0.013$ | 4 | $0.575 \pm 0.017$ | 15 | $0.524 \pm 0.005$ | 12 |
| | TGCL+ResNet | $0.621 \pm 0.009$ | 8 | $0.593 \pm 0.006$ | 9 | $0.562 \pm 0.039$ | 20 |
| PubMed | GCN+ResNet | $0.807 \pm 0.006$ | 3 | $0.783 \pm 0.005$ | 11 | $0.737 \pm 0.002$ | 8 |
| | TGCL+ResNet | $0.820 \pm 0.002$ | 7 | $0.823 \pm 0.005$ | 15 | $0.805 \pm 0.005$ | 20 |
| Reddit | GCN+ResNet | $0.731 \pm 0.010$ | 3 | $0.675 \pm 0.009$ | 3 | $0.626 \pm 0.015$ | 15 |
| | TGCL+ResNet | $0.727 \pm 0.017$ | 3 | $0.725 \pm 0.010$ | 4 | $0.653 \pm 0.008$ | 11 |

GNNs could not collect useful information from the neighbors due to the massive missing values. However, if we increase the number of layers, GNNs are able to gather more information from the $\mathcal{K}$-hop neighbors and capture latent knowledge. To verify this, we conduct the following experiment: we randomly mask $p\%$ attributes in four datasets, gradually increase the number of layers, and report the performance. In this case study, the number of layers is selected from [2, 3, 4, 5, 6, 7, 8, 9, 10, 11, 12, 13, 14, 15, 20, 25, 30, 40, 50, 60] and the base model is GCN. For a fair comparison, we add ResNet (He et al., 2016) to avoid the vanishing gradient issue. We repeat the experiments three times and record the mean accuracy and standard deviation.

Table 3 shows the performance of TGCL as well as the number of layers where the model achieves the best performance (denoted as #L). By observation, we can see that when the missing rate is 25%, 3 layers or 4 layers GCN has enough capability to achieve the best performance in all four datasets and our proposed method only slightly improves the performance. However, when we increase the missing rate to 50% and 75%, we observe that both GCN and TGCL achieve the best performance by stacking more layers and our proposed method improves the performance of GCN by 6.8% in the PubMed dataset when 75% attributes are missing. The experimental results verify that the more values a dataset is missing, the more layers GNNs need to stack to achieve better performance. Our guess for this observation is that if the number of layers increases, more information will be collected from the $\mathcal{K}$-hop neighbors to recover the missing information of its 1-hop and 2-hop neighbors.

## 4.4 EFFICIENCY ANALYSIS

In this subsection, we conduct an efficiency analysis regarding our proposed method on Cora dataset. We fix the feature dimension of the hidden layer to be 50 and we choose GCN as the base model. We gradually increase the number of layers and record the running time. In Figure 2 (c), the x-axis is the number of layers and the y-axis is the running time in second. We could see that the running time of our proposed method is linearly proportional to the number of layers.

# 5 RELATED WORK

In this section, we briefly review the related work on the oversmoothing of GNNs and contrastive learning methods.

## 5.1 OVERSMOOTHING OF GNNS

Oversmoothing problem of GNNs is formally proposed by (Li et al., 2018) by demonstrating that node representations become indiscriminative after stacking many layers in GNN models. This problem is also analyzed by (Oono & Suzuki, 2020) showing how oversmoothing hurts the node classification performance. To quantify the degree of oversmoothing, different measurements are proposed (Chen et al., 2020a; Zhao & Akoglu, 2020; Liu et al., 2020; Zhou et al., 2020). For example, Mean Average Distance (Chen et al., 2020a) is proposed by calculating the divergences between learned node representations. To tackle the oversmoothing problem of GNNs, some nascent research works are proposed (Klicpera et al., 2019; Chen et al., 2020a; Zhao & Akoglu, 2020; Rong et al., 2020; Chen et al., 2020b; Liu et al., 2020; Zhou et al., 2020). They share the same logic of keeping the divergence between node representations but differ in specific methodologies like adding the divergence regularizer in the learning process and changing input graph structures. Taking the union set of these methods' merits, we propose three metrics as shown in Table 1 to comprehensively evaluate a de-oversmoothing method. For example, APPNP (Klicpera et al., 2019) personalizes the information propagation for each specific node to tackle the oversmoothing problem. To be specific, APPNP uses the stationary distribution of random walks to propagate information, which is constant and not changing with the depth of GNN models. However, the number of power iterations to get the approximated stationary distribution is hard to determine and its effect on alleviating the oversmoothing is not clear. Also, in MADReg (Chen et al., 2020a), the divergence regularizer is built on the learned node representation, which is varying with the depth of GNN models, and may not be as robust as the constant divergence indicator. To the best of our knowledge, the proposed TGCL is the first de-oversmoothing method attempting to maintain three metrics at the same time.

## 5.2 CONTRASTIVE LEARNING

Recently, contrastive learning attracts researchers' great attention due to its prominent performance for unsupervised data. (van den Oord et al., 2018) is one of the earliest works, which proposes a Contrastive Predictive Coding framework to extract useful information from high dimensional data with a theoretical guarantee. Based on this work, recent studies (Song & Ermon, 2020; Chuang et al., 2020; Khosla et al., 2020; Tian et al., 2020; Chen et al., 2020c) reveal a surge of research interest in contrastive learning. (You et al., 2020) propose a graph contrastive learning (GraphCL) framework utilize different types of augmentations method to incorporate various priors and to learn unsupervised representations of graph data. (Qiu et al., 2020) propose a Graph Contrastive pre-training model named GCC to capture the graph topological properties across multiple networks by utilizing contrastive learning to learn the intrinsic and transferable structural representations. (Hassani & Ahmadi, 2020) aims to learn node and graph level representations by contrasting structural views of graphs. In this paper, we leverage the topological structure information to contrast the node representations to maximize the similarity of two connected nodes and to minimize the similarity of two remote nodes.

# 6 CONCLUSION

In this paper, we first survey the current de-oversmoothing methods and take the union of their own merits to propose three metrics to evaluate a de-oversmoothing method, i.e., constant divergence indicator, easy-to-determine divergence indicator, alleviating-oversmoothing strategy, and model-agnostic strategy. Then, we propose the Topology-guided Graph Contrastive Layer, named TGCL, which is the first de-oversmoothing method maintaining the three mentioned metrics. With the contrastive learning manner, we provide the theoretical proof of our proposed TGCL and demonstrate the effectiveness of the proposed method by extensive experiments comparing with state-of-the-art de-oversmoothing algorithms.

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
