# OpenReview forum: "Tackling Oversmoothing of GNNs with Contrastive Learning"
_ICLR.cc/2022/Conference — ICLR 2022 Submitted_

### Official Review · Reviewer_ax27 · 2021-10-20

**Correctness:** 3
**Technical Novelty And Significance:** 2
**Empirical Novelty And Significance:** 2
**Recommendation:** 3
**Confidence:** 4

**Main Review:**

Strengths:

Over-smoothing has been an essential topic in GNN and seriously limiting the development of deep models. In this paper, a summary of the current techniques tackling over-smoothing on GNN is reduced, which could be beneficial for insightful GNN variants design in this community.

The proposed TGCL is a flexible plugin to include contrastive information on graphs, and the experimental results somehow evaluate the effectiveness of the proposal.

Weakness:

1.    The literature review of current works tackling over-smoothing is not sufficient, e.g., different architectures of Skip Connections [1], subgraph sampling [2], and DropNode [3]. And the matrices are not clearly explained, studied, or followed in the following content. The formulations of these compared methods need to be illustrated.
2.    The discussion of current works is coarse, making the claimed contribution not grounded and the motivation of this work fuzzy.
3.    The theoretical analysis of TGCL is not consistent with the experiments. For example, the graph sparsity is not studied as discussed in Lemma 1. And the computation cost of TGCL should be high, which should be compared.
4.    The experiments are not well designed since the basic comparisons of the nascent methods are missing, and the "repeated 5 times" seems unreliable. The results are trivial.
5.    The overall homophily should be explicitly claimed in the preliminaries or background.

Ref:

[1] Xu, Kaidi, et al. "Topology attack and defense for graph neural networks: An optimization perspective." arXiv preprint arXiv:1906.04214 (2019).

[2] Zeng, Hanqing, et al. "Deep Graph Neural Networks with Shallow Subgraph Samplers." arXiv preprint arXiv:2012.01380(2020).

[3] Feng W, Zhang J, Dong Y, et al. Graph Random Neural Network for Semi-Supervised Learning on Graphs[J]. arXiv preprint arXiv:2005.11079, 2020.


**Summary Of The Paper:**

This paper summarizes the current techniques tackling over-smoothing, where three matrices are first introduced to describe better and classify the techniques' characteristics. To this end, TGCL, a model-agnostic regularization term based on contrastive learning, is proposed to deal with over-smoothing, satisfying the three defining characteristics.

**Summary Of The Review:**

Based on the weaknesses of the paper pointed out above, I tend to reject this paper.

---

### Official Review · Reviewer_R5AE · 2021-10-23

**Correctness:** 2
**Technical Novelty And Significance:** 1
**Empirical Novelty And Significance:** 1
**Recommendation:** 3
**Confidence:** 5

**Main Review:**

Strengths:
1. The proposed layer TGCL is model-agnostic.

2. The paper is easy to understand.

Weakness:
1. Incorrect math derivations for upper bound. The inequality after Eq.(3) is incorrect. The term
$\sum_{v_i \in N}\text{log}(\sum_{v_k \notin N_i}e^{-||z_i-z_k||^2})$ can be negative. Thus,
$\sum_{v_i \in N}\text{log}(\sum_{v_k \notin N_i}e^{-||z_i-z_k||^2})$ can be larger than  $\sum_{v_i \in N}\sum_{v_j \in N_i}\text{log}(\sum_{v_k \notin N_i}e^{-||z_i-z_k||^2})$ and the $\leqslant$ does not hold. In this sense, the contrastive loss is not the upper bound.

2. Non-rigorous/wrong math derivations for lower bound. In page 6, the proof for your Lemma 1, you use $\approx$ in the derivations, which is not acceptable to derive a lower bound.

3. No formal/math definition for "constant divergence indicator" and "easy-to-determine divergence indicator"

4. The paper fails to explain why "constant divergence indicator" and "easy-to-determine divergence indicator" are a "must" for GNNs to overcome oversmoothing

5. Fig.2 (a) only provides plots for weak baselines. GCNII and drop edge should perform much better.

**Summary Of The Paper:**

The paper proposes a new layer TGCL for GNNs which is used to tackle the oversmoothing problem. This model satisfies three properties: constant divergence indicator, easy-to-determine divergence indicator, and model-agnostic strategy. They show their model alleviates the oversmoothing problem through experiments on 4 datasets.

**Summary Of The Review:**

Overall, based on the weakness mentioned above, I think this paper has severe problems with math derivations and the contribution of this paper is also unclear.

---

### Official Review · Reviewer_5qKE · 2021-11-01

**Correctness:** 3
**Technical Novelty And Significance:** 2
**Empirical Novelty And Significance:** 2
**Recommendation:** 5
**Confidence:** 4

**Main Review:**

Strengths
i)	It surveys most of the existing works and provide three metrics to evaluate the oversmoothing problem in GNN.
ii)	The paper is well-written and easy to follow.
iii)	It provides the theoretical analysis of the proposed loss to show the soundness.

Weakness
i)	As there are lots of existing works such as APPNP, PariNorm, DropEdge etc. Why do authors focus on PairNorm to analyze boundaries? I cannot find the motivation that why choose PairNorm.
ii)	The novelty is limited. The authors analyze the boundary of PairNorm and propose the TGCL. 	Where the majority contribution of this paper is the topology-guided contrastive loss. It is not clear that the meaning of Lemma 1. Why do we need such a lemma? I cannot find any relation between this lemma and the main content.
iii)	Table 2 shows that the improvements are trivial. Compare with the results in GCNII, TGCL does not outperform it in all the datasets. Moreover, GCNII is outperformed TGCL in CiteSeer, and more stable in all those four datasets than TGCL. For example, GCNII got 0.7179 ± 0.0012 in Cora, TGCL got 0.7199 ± 0.0151. In my vision, TGCL only achieves 0.0002 improvements but introduces larger std to make the result unstable.
iv)	Why there are no experiments that use GCNII or other methods in Table 3?



**Summary Of The Paper:**

This paper proposes a method named TGCL which aims to deal with the over smoothing problem in GNN when a number of layers go deeper. Different from existing works, the proposed TGCL considers three different concepts which are Constant
Divergence Indicator, Easy-to-Determine Divergence Indicator, and Model-Agnostic Strategy. Most of the existing works are only considered two of them. The proposed method can be recognized as an improvement of the PairNorm.


**Summary Of The Review:**

Overall, this paper provides a new way to define the oversmoothing problem in GNN. But it lacks the novelty and solid experiments.

---

### Official Review · Reviewer_wVaG · 2021-11-03

**Correctness:** 3
**Technical Novelty And Significance:** 3
**Empirical Novelty And Significance:** 3
**Recommendation:** 6
**Confidence:** 4

**Main Review:**

Pros:
- the manuscript is well organized and easy to follow;
- good analysis on K-hop study in section 4.3, which explains the intuition of the topology guided method;
- the method is generic and easy to add to baseline GNN models which could serve as a contrastive loss extension;

Cons:
- the three metrics that the author proposed to compare with existing methods are marginal put: most of the baseline model satisfied the third metric(model-agnostic) and as for the second metric: the TGCL method is claimed to be easy to adopt but the loss term definition includes extra parameters like temperature, choice of distance method, similarity function, alpha, etc. The experiment section didn't expand on this to show how the optimal configurations are chosen, given the margin of improvement in Table.2 these parameters might need to be analyzed;
- Convergence speed compared to baseline methods are not mentioned in the paper especially when loss term is added in the learning process;

**Summary Of The Paper:**

In this work, the author analyzed the oversmoothing issue in GNN learning. To compare with the related works, the author proposed three metrics: divergence indicator, the intuition to set its value, and the ease to adopt. The author proposed a new method Topology-guided Graph Conttrasive Layer which achieved de-oversmoothing and maintaining the proposed metrics. Lower bound analysis of the TGCL loss was analyzed and the empirical experiment showed an advantage on four graph datasets.

**Summary Of The Review:**

The paper is well written and the method proposed is marginally novel. An extra support section for optimal configuration and convergence analysis would be a great plus to justify the advantage of the method.

---

### Decision · Program_Chairs · 2022-01-20

**Decision:**

Reject

**Comment:**

This paper introduces a new layer for graph neural networks that aims to reduce the oversmoothing issue common to this model type. The reviewers find the paper well organized and easy to follow, and they recognize the importance of the problem that is addressed. However, they also identify critical errors in the mathematical derivations: the authors did not provide a response to the reviews, and hence these errors remain unaddressed. In addition, multiple reviewers indicate they find the experimental evaluation insufficient. For these reasons I'm recommending rejecting this paper.